# Serum and Soleus Metabolomics Signature of *Klf10* Knockout Mice to Identify Potential Biomarkers

**DOI:** 10.3390/metabo12060556

**Published:** 2022-06-17

**Authors:** Nadine Baroukh, Nathan Canteleux, Antoine Lefèvre, Camille Dupuy, Cécile Martias, Antoine Presset, Malayannan Subramaniam, John R. Hawse, Patrick Emond, Philippe Pouletaut, Sandrine Morandat, Sabine F. Bensamoun, Lydie Nadal-Desbarats

**Affiliations:** 1UMR 1253, iBrain, University of Tours, Inserm, 37044 Tours, France; nadine.baroukh@univ-tours.fr (N.B.); canteleux@gmail.com (N.C.); antoine.lefevre@univ-tours.fr (A.L.); camille.dupuy@allice.fr (C.D.); cmartias@pasteur-guadeloupe.fr (C.M.); antoine.presset@univ-tours.fr (A.P.); patrick.emond@univ-tours.fr (P.E.); 2Department of Biochemistry and Molecular Biology, Mayo Clinic College of Medicine, Rochester, MN 55905, USA; subramaniam.malayannan@mayo.edu (M.S.); hawse.john@mayo.edu (J.R.H.); 3CHRU Tours, Medical Biology Center, 37000 Tours, France; 4Biomechanics and Bioengineering Laboratory, CNRS UMR 7338, Université de Technologie de Compiègne, 60200 Compiègne, France; philippe.pouletaut@utc.fr (P.P.); sandrine.morandat@utc.fr (S.M.); sabine.bensamoun@utc.fr (S.F.B.)

**Keywords:** *Klf10*, metabolomics, metabolic pathways, Warburg effect, mice, soleus, serum, UHPLC-MS

## Abstract

The transcription factor Krüppel-like factor 10 (*Klf10*), also known as Tieg1 for TGFβ (Inducible Early Gene-1) is known to control numerous genes in many cell types that are involved in various key biological processes (differentiation, proliferation, apoptosis, inflammation), including cell metabolism and human disease. In skeletal muscle, particularly in the soleus, deletion of the *Klf10* gene (*Klf10* KO) resulted in ultrastructure fiber disorganization and mitochondrial metabolism deficiencies, characterized by muscular hypertrophy. To determine the metabolic profile related to loss of *Klf10* expression, we analyzed blood and soleus tissue using UHPLC-Mass Spectrometry. Metabolomics analyses on both serum and soleus revealed profound differences between wild-type (WT) and KO animals. *Klf10* deficient mice exhibited alterations in metabolites associated with energetic metabolism. Additionally, chemical classes of aromatic and amino-acid compounds were disrupted, together with Krebs cycle intermediates, lipids and phospholipids. From variable importance in projection (VIP) analyses, the Warburg effect, citric acid cycle, gluconeogenesis and transfer of acetyl groups into mitochondria appeared to be possible pathways involved in the metabolic alterations observed in *Klf10* KO mice. These studies have revealed essential roles for *Klf10* in regulating multiple metabolic pathways whose alterations may underlie the observed skeletal muscle defects as well as other diseases.

## 1. Introduction

The transcription factor Krüppel-like factor 10 (*Klf10*), also named TGF-β (Trans-forming Growth Factor) inducible early gene 1 (*Tieg1*), is part of a complex and specific regulatory transcription system that regulates numerous genes in a variety of cells and tissue types [1,2,3]. Alterations in the expression of *Klf10* have implications on multiple biological processes (proliferation, differentiation, apoptosis, inflammation) [4], relevant to disease including diabetes, obesity, cataracts, cardiac hypertrophy, and angiogenesis, as well as the development and progression of different cancer types (breast, kidney, pancreas, prostate, and ovarian) [5,6]. *Klf10* is ubiquitously expressed in mammals and exhibits a higher expression in metabolic organs such as the liver, pancreas, muscle, and adipose tissue. *Klf10* controls a variety of biological processes in the liver and skeletal muscle. It plays a crucial role in the regulation of the circadian clock, driving the expression of genes involved in lipogenesis, gluconeogenesis (with a higher fasting glucose), and glycolysis in the liver, influencing hepatic metabolism [7,8]. Moreover, circadian perturbation may initiate metabolic disorders and tumorigenesis. In this context, it has been shown that *Klf10* is downregulated in type 2 diabetes and cancer [9]. For example, *Klf10* expression is diminished during breast cancer evolution [1,10]. *Klf10* is described as a tumor suppressor gene through TGF-β-induced growth inhibition; its functions are anti-proliferative and pro-apoptotic in cancer cells [11,12,13,14]. More recently, *Klf10* has been described to be protective against oral cancer [15] and has been implicated in immune CD4+ T-cell localization in tissues, leading to obesity, insulin resistance, and fatty liver development [16]. Data indicate that KLF10 acts as a central pivot, coordinating circadian rhythm together with metabolic pathways and energy metabolism homeostasis [3,17,18].

The generation of *Klf10* knockout (KO) mice [19] has demonstrated its involvement in musculoskeletal tissue deficiencies such as defects in structure and healing of tendons [20,21], hypertrophic cardiomyopathy [2], and bone diseases (osteopenia, osteoporosis) [19,22,23,24,25]. In mice, *Klf10* gene deletion results in hyperplasia and structural changes such as hypertrophy and ultrastructure disorganization in the soleus (slow-twitch muscle) and extensor digitorum longus (fast-twitch muscle). It has been shown that *Klf10* deletion induces changes in passive [26] and active [27] behaviors in a fiber type-specific manner. *Klf10* deficient mice have contributed to our understanding of skeletal muscle metabolism, which is central to overall energetic metabolism in the animal. *Klf10* is known to be responsible for important mitochondria regulating functions, affecting succinate dehydrogenase (SDH), cytochrome C oxidase (COX), and the citrate synthase (CS) enzyme activities in the soleus of deficient mice [28].

The metabolome refers to the representation of small chemical molecules (or metabolites) contained in a given biological sample at a specific time. In a biological sense, the metabolome represents a report of multiple cellular processes, including the regulation of the genome, transcriptome, and proteome cascades, and ultimately supports phenotypes and functions. Despite the limit point being the lack of insights between the production and consumption of metabolites, metabolomics has become a powerful tool to better understand the physiology, pathophysiology, or dysregulation of many biological systems [29,30,31]. Blood (or serum) is frequently used in metabolomics research as well. Indeed, it contains many metabolites from the entire body and gives a global and integrative view of what is occurring at a specific time. Metabolites are known to be involved in a particular biological context, help shed light on alterations in associated metabolic pathways, and can serve as specific biomarkers for a variety of conditions and diseases.

The impact of *Klf10* on metabolism remains unclear and must be further investigated. This study is a continuation of our work demonstrating a role for *Klf10* in muscle metabolism [28]. Here, we describe the metabolomic profile of *Klf10* KO mice at the organismal level (serum) and the tissue level (muscle). This work has led to a better understanding of the role of *Klf10* in metabolism, the identification of specific metabolic pathways regulated by *Klf10*, and the identification of potential biomarkers relevant for skeletal muscle defects/diseases.

## 2. Results

### 2.1. Metabolomic Analyses of Serum and Soleus

We aimed to characterize differential metabolomics profiles in the serum and soleus of *Klf10* KO versus wild-type mice. A total of 224 and 246 metabolites were detected in serum and soleus, respectively. The PLS-DA score plot analysis of *Klf10* KO and WT samples of serum and soleus are shown in Figure 1. Results indicate that both serum (Figure 1A) and soleus (Figure 1B) metabolic profiles are distinctly separated between the two groups of mice. The best PLS-DA models for serum retained 96 variables with importance in projection (VIP) with the performance characteristics of R^2^Y = 0.974, Q^2^ = 0.87, CV-ANOVA = 4.42 × 10^−6^, from which 78 variables showed a *p*-value < 0.05. In soleus, the best PLS-DA score plot used 88 VIP with R^2^Y = 0.93, Q^2^ = 0.86, CV-ANOVA = 5.46 × 10^−7^, from which 27 variables have a *p*-value < 0.05. The VIP represent metabolites with a greater contribution to the separation between groups (*Klf10* KO versus WT). The segregation into two distinguishable groups indicates that *Klf10* gene deletion caused significant changes in the overall metabolome in mice. Of these, all metabolites showing a significant change are listed in Appendix A for serum and Appendix A for soleus.

To define relevant metabolites implicated in the clustering between *Klf10* KO and WT mice, we performed a volcano plot analysis (Figure 2) with the former VIP for each biological sample and found 34 metabolites in serum (Figure 2A) and 9 metabolites in soleus (Figure 2B), representing 35.4% and 10.1% of metabolites, respectively, that were highly regulated (FC > 1.5 or FC < 0.5 and *p*-value < 0.05). For instance, cholic acid and glycocholic acid in serum and caprylic acid and cortisol in soleus were highly upregulated in *Klf10* KO compared to WT mice. Interestingly, hippuric acid was highly downregulated in both samples of *Klf10* KO mice. These data demonstrated profound differences in the metabolic profiles of serum (Appendix A) and soleus (Appendix A) of *Klf10* KO mice.

### 2.2. Characterization of Metabolic Changes in Klf10 KO Mice

VIP were classified according to their chemical classes (Figure 3). The percentages of the chemical classes between *Klf10* KO and WT mice are presented in the pie charts for serum (Figure 3A) and for soleus (Figure 3B). This revealed nine different chemical classes. The inter-biological sample (serum vs. soleus) comparison showed that metabolites mainly archived into phospholipids (27% in serum vs. 16% in soleus), amino acids and derivatives (23% in serum vs. 21% in soleus), lipids: fatty acids-steroids (18% in serum vs. 15% in soleus), and aromatic compounds—amines (7% in serum vs. 8% in soleus). Organic acids were similarly modified, exhibiting 10% variation in both serum and soleus between *Klf10* KO and WT mice. A slight difference was noted for sugar (3% in serum vs. 7% in soleus), nucleotides–nucleosides derivatives (6% in serum vs. 14% in soleus), and vitamins—cofactors—hormones (2% in serum vs. 7% in soleus). With the exception of sugar, phospholipids, nucleotides–nucleosides derivatives, and vitamins—cofactors–hormones classes, a very similar perturbation was observed on chemical repartition for serum and soleus, indicating that the same classes of chemical compounds were impacted in both biological samples in *Klf10* KO mice providing strong evidence for direct regulation of these metabolites by *Klf10*.

In serum, numerous metabolite changes were observed in particular in lipid and phospholipid classes, including palmitic acid (FA), phosphatidylcholine (PC), phosphatidylethanolamine (PE), lysophosphocholine (LPC), and platelet-activating factor (PAF) or PC (O-) (Appendix A and Figure 2A). There were a total of 26 phospholipids over the 96 VIP, representing 27% of the overall change in serum. We also found increased levels of plasma lipids and acylcarnitines, such as L-acetylcarnitine, O-acetyl-L-carnitine, deoxycarnitine, tetradecanoylcarnitine, propionylcarnitine, and dodecanoylcarnitine. In addition to lipid metabolism, amino acid metabolism was significantly altered in the serum of *Klf10* KO mice. We found 23% amino acid change, including L-threonine, L-lysine, L-valine, L-serine, L-arginine, L-asparagine, L-methionine, L-ornithine, and derivatives such as N-acetyltryptophan, indicating that amino acid metabolism was clearly impacted. Metabolic characterization in serum identified compounds from glycolysis and the Krebs cycle, including a slight decrease in D-glucose and L-lactic acid and a significant increase in citric acid, isocitric acid, oxoglutaric acid, fumaric acid, and L-malic acid (*p*-value < 0.05). Together, changes in chemical classes of sugars and organic acids represented 3% and 10% of the overall changes, respectively (Figure 3A), suggesting that energetic metabolism (glycolysis and Krebs cycle intermediates) is impacted and reflected in the serum of *Klf10* deficient mice.

In the soleus, phospholipids were less altered than was reflected in the serum (Figure 3). The PC and LPC were predominantly impacted (Appendix A), and acylcarnitines (propionylcarnitine, desoxycarnitine, tetradecanoylcarnitine, which are common with serum, and decanoylcarnitine, which is specific to soleus) followed the same trend. Alterations in metabolites of glycolysis and the Krebs cycle, including glycogen and purine nucleotide, were identified (Figure 4). D-glucose and L-lactic acid were significantly increased in the soleus of *Klf10* KO versus WT mice (*p*-value < 0.05). In the Krebs cycle, citric acid, isocitric acid, succinic acid, fumaric acid, and L-malic acid were detected and demonstrated fluctuations, together with amino acids (L-arginine, L-aspartic acid, and L-methionine). Glucose 1-phosphate (G1P) and glucose 6-phosphate (G6P) were differentially discriminated as VIP and fed the glycogen storage metabolism. Interestingly, nucleotides–nucleosides, derivatives such as guanosine, guanine, inosine, and uric acid that are involved in purine metabolism, exhibited a significant decrease in *Klf10* deficient mice. In this regard, lipids, phospholipids, sugar, amino acids, nucleotides, and organic acids were determined to be altered in the soleus of *Klf10* KO mice (Figure 4).

### 2.3. Identification of Metabolic Pathways Involved in Klf10 KO Mice

Following the metabolic change analyses in serum and soleus, we investigated metabolite set enrichment pathways [32]. It is important to note that lipids and phospholipids are not considered by the software for data treatment to establish enrichment set diagrams. Despite this, we characterized the incidence of change implicating VIP in serum and in soleus on metabolic pathways (Figure 5). Enrichment diagrams revealed that in both samples, metabolite changes affected the Warburg effect (aerobic glycolytic metabolism), the citric acid cycle, the transfer of acetyl group into mitochondria, the ammonia recycling, the arginine and proline metabolism, gluconeogenesis, the thiamine metabolism, the glycine and serine metabolism, the urea cycle, and many other pathways to a lesser extent. Metabolic pathways highlighted by enrichment analyses revealed alterations that affect particularly energetic and amino acid metabolisms and transfer–degradation–recycling processes (Figure 5).

Given these findings and the similarities in terms of influenced metabolic pathways, we generated a Venn diagram to compare VIP obtained from serum and soleus (Figure 6A). Analyzing intersections between the two matrices identified 36 common metabolites, representing 38% of the VIP in serum and 41% in soleus. Common metabolites are listed in Figure 6, and their chemical repartition is depicted in the pie chart (Figure 6B). As expected, we found metabolites involved in energetic metabolism (carbohydrate and lipid), such as D-glucose, some phosphatidylcholines (PC) and lysoPC, carnitine derivative compounds, and Krebs cycle intermediates (citric acid, fumaric acid, isocitric acid, and malic acid). Other common metabolites included amino acids, identifying L-arginine, ketoleucine, L-methionine, L-serine, L-threonine, or derived degradation products such as hippuric acid, serotonin, and urocanic acid. To focus on the commonly altered pathways in both biological samples, an overview of Metabolic Set Enrichment Analysis (MSEA) was run using the 36 metabolites shared by the biological samples (Figure 6C). This identified four noteworthy metabolic pathways; the Warburg effect, the citric acid cycle (Krebs cycle), the gluconeogenesis, and the transfer of acetyl groups into mitochondria. By extracting information from each pathway, it was possible to visualize that D-glucose, lactic acid, isocitric acid, and citric acid were upregulated in the soleus (Figure 2B), and biotin, fumaric acid, and malic acid were downregulated in the soleus (Figure 2B). These represent highly relevant metabolites impacted by *Klf10* in both compartments (Figure 6).

As a next step, we compared the fold change of the 36 common metabolites found in serum and soleus using a radar diagram (Figure 7). This representation allowed for the visualization of compounds having similar or different behaviors between the biological samples. For example, orotic acid, palmitic acid, some PC (34:1; 36:2 and O-14:0/2:0), uracil, uric acid, and uridine were impacted similarly by *Klf10* loss while D-glucose, biotin, serotonin, tretradecanoylcarnitine, LPC(18:0), L-arginine, deoxycarnitine, thiamine monophosphate, platelet-activating factor, PC(O-16:1/2:2), and N-methyltryptamine were oppositely impacted (see Appendix A).

### 2.4. Identification of Metabolic Pathways Involved in Klf10 KO Mice

Our results clearly demonstrated a specific signature of change in the metabolomic profile of *Klf10* KO mice. The drastic metabolomics differences observed reliably separated the two groups of mice (*Klf10* KO vs. WT mice) and allowed for the identification of potential biomarkers that are reflective of decreased *Klf10* expression and/or function. To highlight potential biomarkers of interest in serum and soleus, we applied a strict cut-off of two-fold change and *p*-value < 0.05 (Figure 8).

In serum, we found a set of 17 metabolites highly regulated, which mostly belong to lipids, phospholipids, and amino acids—derivatives and aromatic compounds—amines classes. In a global picture, we observed that metabolites from lipids and phospholipid classes (PE(22:6/18:1), PE(20:4/18:1), PE(16:0/20:4), tetradecanoylcarnitine, glycocholic, acid, and cholic acid were upregulated whereas metabolites from aromatic compounds-amines class (indole-3-methyl acetate, N-methyltryptamine, and trigonelline) and amino acids-derivative compounds (ketoleucine, N-acetylserotonin, serotonin, hippuric acid) were downregulated. Some others, such as indoleacetic acid and biotin, were upregulated. In serum, guanidoacetic acid and N-acetyltryptophan from amino acid-derivative groups were also upregulated. All dysregulated metabolites (FC > 2 and *p*-value < 0.05) described here may serve as potential robust biomarker predictors for lack of *Klf10* expression and/or function.

In the soleus, we identified nine metabolites exhibiting a two-fold change but only three with a *p*-value < 0.05. Urocanic acid, caprylic acid, and hippuric acid were highly dysregulated in soleus and belonged to the chemical classes of lipid: fatty acids-steroids (caprylic acid, upregulated) and amino acids–peptides derivatives (hippuric acid and urocanic acid, downregulated).

## 3. Discussion

In this study, using UHPLC-Mass Spectrometry, we conducted a serum and soleus metabolic analysis to establish a specific metabolic signature of *Klf10* KO mice. Indeed, the analysis of both serum and soleus in *Klf10* KO mice succeeded in detecting a total of 224 and 246 altered metabolites, respectively. The PLS-DA model for serum revealed 96 variables with importance in projection (VIP) and 88 VIP for the soleus, which contributed to establishing the models with robust values. We found a drastic change in metabolites that clearly separated *Klf10* KO from wild-type mice in each biological sample. We noticed an increased number of metabolites significantly regulated in serum when compared to the soleus, and the overall data demonstrated a profound metabolome alteration in *Klf10* KO compared to WT mice.

Metabolomic profiles found in *Klf10* KO mice affected different categories of compounds that belong to various chemical classes, such as lipids, phospholipids, amino acids, aromatic compounds, vitamins, hormones, and others. The proportion diagram of compound class repartition is mostly similar between serum and soleus, except for sugar, phospholipids, nucleotides-nucleosides derivatives, and vitamins–cofactors–hormones classes. This suggests that *Klf10* deficiency impacts carbohydrate, nucleic acid, and lipid metabolism differently in respect of these two samples. In this context, results indicate a metabolic change in (1) fatty acid metabolism driven by phospholipids, lipids, and acylcarnitine modifications, (2) amino acid metabolism by glucogenic amino acid fluctuations, and (3) energetic metabolism related to glycolysis and Krebs cycle metabolite alterations. Changes in serum fatty acid and acylcarnitine levels could have a high predictive value for segregating *Klf10* KO and WT mice, as shown by Houtlooper et al. in plasma [33]. Interestingly, plasma acylcarnitine elevation suggests the transfer of acetyl groups into the mitochondria for energy production via lipid catabolism. It is worthy to note that increases in acylcarnitine are found in the pathologies of obesity and diabetes (type II) [34,35]. In soleus specifically, the elevated levels of glucose, glucose-1-phosphate, and glucose-6-phosphate suggest changes in glycolysis and glycogen metabolism, while guanine, inosine, and uric acid affect the metabolism of purine nucleotides. Thus, in agreement with the literature, there is increasing evidence in favor of KLF members as transcriptional regulators of energy metabolism [17]. In summary, metabolic results obtained in this study argue in this direction and give a representative window of the metabolic state of *Klf10* KO mice (Figure 4). In *Klf10* KO mice, we observed a drastic metabolome reorganization affecting mainly lipids, phospholipids, amino acids, and to a lesser degree, nucleotides.

To expand our analysis, we linked changes in VIP metabolites identified with metabolic pathways. We discovered that the Warburg effect and the citric acid cycle were among the most highly affected pathways in both the soleus and serum (Figure 5 and Figure 6). Interestingly, the Warburg effect is an aerobic glycolysis metabolism that occurs mainly in cancer and proliferating cells. This particular type of metabolism improves nutrient uptake to directly incorporate biomass, generating nucleotides, amino acids, and lipids. It is a process that induces cells to acquire proliferation potential rather than using nutrients for energy production [36].

Our previous study demonstrated that *Klf10* KO mice exhibit disruption in the Krebs cycle and oxidative phosphorylation in mitochondria by acting on citrate synthase (CS) and succinate dehydrogenase (SDH, complex II) enzymes [28]. Based on the present discovery, we presented a hypothetical metabolic model of change occurring in the soleus of *Klf10* KO mice in Figure 9, where the flux of glucose through glycolysis is slowed down at the mitochondria level (caused by CS and SDH impairment). As a consequence, it is expected that a small fraction of pyruvate enters into the Krebs cycle, and the rest is highly converted into lactic acid, which may accumulate and serve to produce energy by anaerobia. At that point, citric acid and isocitric acid are blocked at the Krebs cycle level, and the excess citric acid may follow the lipid synthesis pathway through acetyl CoA conversion. In parallel, glycolysis intermediate products may accumulate and be reoriented to amino acid synthesis and/or glycogen storage. It appears that the soleus of *Klf10* KO mice reproduces Warburg metabolism that is related to the functions of KLF10 described in the pathogenesis of various cancers [37]. Further research is needed to fully establish these possibilities and validate the proposed model.

Aging is a late-life state characterized by a general degradation of cell function leading to declines in health. Increasing evidence shows that mitochondrial dysfunctions contribute to the aging process [38,39], including insulin signaling pathways, which are known to affect longevity [40,41]. Further, mitochondria are required and play a crucial role in skeletal muscle insulin signaling [42]. As discussed above, KLF10 affects metabolic functions with strong involvement in impaired mitochondrial functions [28]. In addition, as a glucose-responsive gene in the liver [7], KLF10 plays a prominent role in regulating insulin signaling pathways that drive Pi3k-Akt for glycogenesis and mTOR (mammalian target of rapamycin) signaling for protein synthesis [16,43,44]. One of the aspects of regulating the insulin signaling pathway results in glucose and lipid homeostasis, together with protein synthesis via mTOR. In our study, we observed that *Klf10* deletion in mice induced alterations in a large panel of molecules implicated in the regulation of cell activities at different levels (energetic metabolism, molecular, and signaling processes, including hormones and neurotransmitters). We proceeded to compare the metabolic signatures of *Klf10* KO mice to specific metabolic profiles characterized in aged mice, where amino acids, glucose, and lipid metabolism are affected in both liver and muscle [33], as well as in adult human plasma [45]. In a more precise manner, linoleic acid decreases have been shown to be a robust biomarker of aging [33]. We also observed a decrease in linoleic acid in *Klf10* KO serum (Appendix A). Similarly, we noticed that metabolic products associated with phospholipid metabolism are significantly changed in the soleus of *Klf10* KO mice; this is also described in the skeletal muscle of aged mice that seems to reflect membrane cell modifications and altered functions [46,47]. Importantly, phospholipids take part in molecular transduction signaling for gene regulation, as well. In the soleus of *Klf10* KO mice, glycogen metabolism is altered by increased levels of G6P and G1P, while increases in L-lactic acid and glucose levels suggest a Warburg effect (anaerobic glycolysis). A similar situation is recalled in muscle of aging mice, where some metabolic changes implicate glycogen metabolism and anaerobic glycolysis, leading to age-related insulin resistance profiles [33,48]. Comparably, changes in other distinct metabolites such as citric acid or serotonin neurotransmitter were increased in the skeletal muscle of aged mice [47]; the last of which was also found to be increased in the soleus of *Klf10* KO mice. These observations provide strong evidence that *Klf10* KO mice share features with known aging phenotypes in skeletal muscle, particularly with regard to impaired mitochondrial function, metabolome footprint, and insulin (and mTOR) signal pathway regulation. In summary, *Klf10* KO mice refer to the deterioration of skeletal muscle cell structure and organization that coincide with biological functions altered with age.

In addition to metabolome characterization, another important goal is to identify specific metabolic biomarkers of *Klf10* deletion. Extensive analysis of the metabolome in serum and soleus allowed us to identify metabolite groups associated with *Klf10* deficiency. At a high level, lipids and amino acids were highly disrupted classes of metabolites altered in both the serum and soleus of *Klf10* KO mice. Some metabolites appeared highly discriminatory and, as such, may represent ideal biomarkers. Present in both matrices, hippuric acid, a derived amino acid, was highly decreased in serum and soleus. Tracking this metabolite in real-time could provide additional information about compartment exchanges and pathophysiology. Additionally, some metabolites were sample-specific such as cholic acid and glycocholic acid that were altered in serum, and caprylic acid and urocanic acid that were altered in the soleus. Among these sets of interesting candidate biomarkers (Figure 8), cholic acid and glycocholic (derivatives of cholic acid) are the major conjugated bile acids that facilitate lipid absorption. When found at high levels, cholic and glycocholic acids provoke hepatic and metabolic toxicities leading to possible membrane disruption. It can be assumed that KLF10 may regulate bile acid synthesis and affect the lipids and phospholipid levels. A similar function has been described for KLF15, a member of the KLF family [49]. One may hypothesize that the lack of *Klf10* leads to a global toxicity profile, implicating *Klf10* as a protective factor against cellular, tissue, and/or organ injury [50,51,52].

## 4. Materials and Methods

### 4.1. Animals

The generation of *Klf10* KO mice has been previously described [19]. To be consistent with our previous studies, we utilized 3-month-old littermate female animals derived from heterozygous breedings. Female mice were chosen based on preliminary results indicating higher expression of *Klf10* in female mice compared to male animals. All mice were maintained in a temperature-controlled room (22 ± 2 °C) with a light/dark cycle of 12 h. Animals had free access to water and were fed with standard laboratory chow ad libitum. The protocol was approved by the French ministry of higher education, research and innovation (Permit Number: DUO-4776) and the local ethics committee Comité Régional d’Ethique en Matière d’Expérimentation Animale de Picardie (CREMEAP; Permit Number: APAFIS #8905-2021011109249708).

### 4.2. Sample Collection and Preparation

Slow twitch-muscle (soleus) was utilized given the defects observed in this glycolytic muscle following the deletion of *Klf10* and the 4-fold higher *Klf10* expression in soleus compared to EDL (28). Thus, the soleus of 3-month-old female mice were isolated from *Klf10* KO (n = 10) and WT (n = 9) mice, immediately frozen, and subsequently stored at −80 °C until metabolomics analysis. For metabolite extraction, 1 mL of MeOH:H_2_O (50:50) was added to frozen muscle samples and agitated for 30 min at 4 °C followed by centrifugation at 5000× *g* for 15 min at 4 °C. In total, 900 µL of supernatants were collected for solvent evaporation in a SpeedVac (ThermoFisher, Villebon sur Yvette, France) at 35 °C for 2 h. The remaining supernatants were pooled for quality control (QC) samples and evaporated.

Serum samples from *Klf10* KO (n = 9) and WT (n = 10) mice were prepared as previously described [53]. Approximately 50 µL of serum were added to 400 µL of MeOH, centrifuged, and the supernatant was collected before evaporation. QC samples (10 µL of each sample) were also prepared.

For UHPLC-MS, evaporated samples were solubilized with 70 µL of solvent MeOH:H_2_O (1:9) for RP-LC columns and 70 µL of ACN:H_2_O (9:1) for HILIC columns.

### 4.3. Ultra-High-Performance Liquid Chromatography-Mass Spectroscopy (UHPLC-MS)

#### 4.3.1. Data Acquisition

As previously described [54], UHPLC-MS analysis was performed on a UHPLC Ultimate WPS-3000 system (Dionex, Idstein, Germany), coupled with a QExactive mass spectrometer (Thermo Fisher Scientific, Bremen, Germany).

The chromatography system was equipped separately with two columns: an RP-LC (Reverse Phase Liquid Chromatography) Phenomenex Kinetex^®^ XB-C18 (1.7 µm 100 A 150 × 2.1 mm from Phenomenex, Torrance, CA, USA) and a HILIC (Hydrophilic Interaction LIquid Chromatography) Waters Cortecs^®^ (unbonded silica; 1.6 µm 100 A 150 × 2.1 mm from Waters, Dublin, Ireland).

The autosampler temperature (Ultimate WPS-3000 UHPLC system, Dionex, Germany) was set at 4 °C.

A HESI (Head ElectroSpray Ionization) source was used for both chromatography systems operated in positive (ESI+) and negative (ESI−) electrospray ionization modes (one run for each mode).

Detection was performed with a full-scan acquisition at 70,000 resolution (*m*/*z* = 200), which ranged from 58.0 to 870.0 *m*/*z*, with an automatic gain control target of 105 charges and a maximum injection time (IT) of 250 ms. Xcalibur 2.2 software (Thermo Fisher Scientific, Bremen, Germany) controlled the system.

#### 4.3.2. Data Processing

The acquired data were processed using Xcalibur^®^ software (Thermo Fisher Scientific, San Jose, CA, USA) by integrating selected product ion chromatographic peak areas, which were exported to an excel file containing the areas of each metabolite and finally normalized to the total sum of integrated metabolites. Detected peaks were identified using accurate *m*/*z* and retention time (RT) compared to the internal database made by the Mass Spectrometry Metabolite Library of Standards (MSMLS^®^, IROA Technologies™, Bolton, MA, USA). According to Rathato-Paris et al. [55], our annotation is considered putative metabolite identification (level 2). Only metabolites detected in quality control (QC) samples with a coefficient of variation (CV) <30% were kept for further analysis.

#### 4.3.3. Data Analysis

Once the RP+/− (positive or negative ionization)-LC and HILIC+/−-LC lists were generated, the web interface Metaboanalyst (https://www.metaboanalyst.ca, accessed on 20 September 2021) [32] was used to standardize the generic name and to generate Human Metabolome Data Base (HMDB) or Kyoto Encyclopedia Genes and Genomes (KEGG) numbers for each metabolite. A fusion step was used to merge the four lists according to their HMDB or KEGG numbers. When metabolites were present in several lists, the platform having the lower CV (for reproducibility on QC samples) was kept for further analysis.

A Venn diagram (http://jvenn.toulouse.inra.fr/app/example.html, accessed on 20 September 2021) [56] was used to identify the common metabolites between serum and soleus.

### 4.4. Statistical Analysis

#### 4.4.1. Multivariate Analysis

In the first intention, a principal component data analysis (PCA) was performed in order to observe and exclude outlier samples. Once these outliers were excluded, a supervised multivariate statistical data analysis based on PLS-DA (Partial Least Squares Discriminant Analysis) was then performed on the metabolomics data generated by RP and HILIC-LC using SIMCA^®^15 software (Umetrics, Umea, Sweden). The model qualities were evaluated by R2Y (goodness of fit), Q2 (goodness of prediction), and CV-ANOVA (Cross Validation-ANalysis Of VAriance). The PLS-DA models were improved according to the regression coefficients of each metabolite that expressed how strongly the Y (WT or *Klf10* KO) is related to the X-variables (metabolites). According to these criteria, metabolites (variable with importance in projection = VIP) with a greater contribution to the separation of the groups were identified.

#### 4.4.2. Univariate Analysis

Univariate analysis was performed on metabolites identified by the PLS-DA using a t-test on Metaboanalyst (https://www.metaboanalyst.ca, accessed on 20 September 2021). The results were considered to be significantly different when the *p*-value was ≤ 0.05. The fold changes were calculated.

#### 4.4.3. Metabolites Set Enrichment

The metabolites implicated in the clustering on the PLS-DA analysis were loaded to identify the most significantly affected metabolic pathways by metabolite-set enrichment analysis (MSEA) as previously described by Beauclercq S. et al. [57]. A cut-off was chosen to present the first 25 metabolic pathways corresponding to the most significantly enriched.

## 5. Conclusions

In conclusion, the results of the studies presented here support that KLF10is highly related to the dynamic processes of metabolism regulation that manage energy (glucose and lipids) production, which are essential to cellular physiology. Considering KLF10′s involvement in gene expression, signal transduction, and metabolism [16], it has significant potential to become a biological cell sensor. It is clear that metabolomics has helped to establish characterized footprints of phenotypes resulting from many interconnected biological mechanisms [58]. In this case, characterizing metabolic profiles and identifying potential metabolic biomarkers are of therapeutic interest and may serve to understand metabolic homeostasis related to disease development and/or various (patho)physiological conditions.

## Figures and Tables

**Figure 1 metabolites-12-00556-f001:**
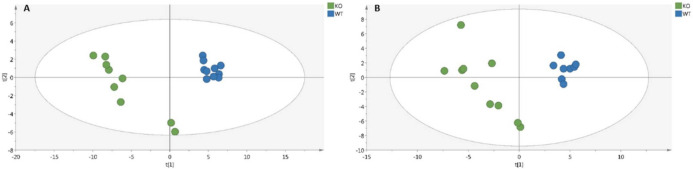
PLS-DA score plot analysis of serum (**A**) and soleus (**B**) samples in *Klf10* KO and WT by UHPLC-MS. (**A**) Metabolome score plots of PLS-DA on serum (WT in blue circle, n = 10 and *Klf10* KO green circle, n = 9). The performance characteristics of the model using 96 variables, are R^2^Y = 0.974, Q^2^ = 0.87 and CV-ANOVA = 4.42 × 10^−6^. (**B**) Metabolome score plots of PLS-DA on soleus (WT in blue circle, n = 9 and *Klf10* KO in green circle, n = 10). The performance characteristics of the model using 88 variables, are R^2^Y = 0.93, Q^2^ = 0.86 and CV-ANOVA = 5.46 × 10^−7^.

**Figure 2 metabolites-12-00556-f002:**
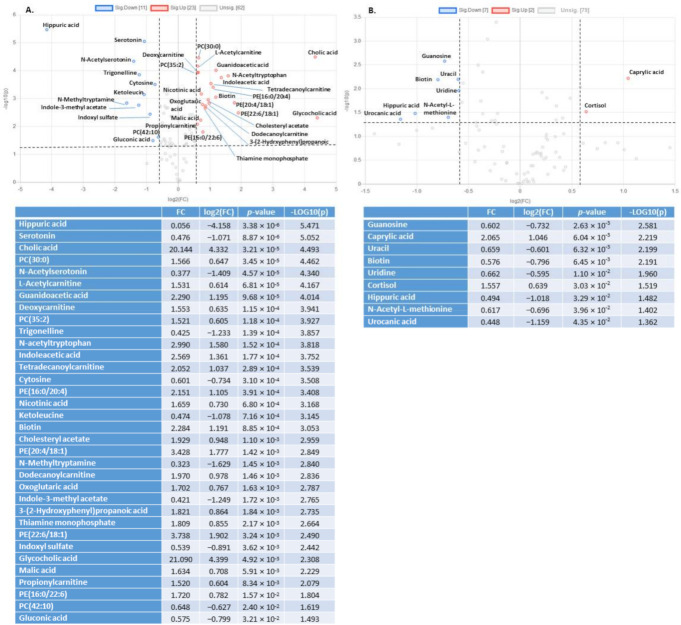
Volcano plots showing metabolites changes in serum (**A**) and soleus (**B**) in *Klf10* KO versus WT. Each point corresponds to a metabolite (as VIP). Significant selected metabolites are upregulated in red and downregulated in blue. In grey, metabolites with 0.5 < FC < 1.5 and/or *p*-value > 0.05. Dotted horizontal line indicates threshold for *p*-value of 0.05 and the dotted vertical line indicates threshold for 1.5 fold change (FC).

**Figure 3 metabolites-12-00556-f003:**
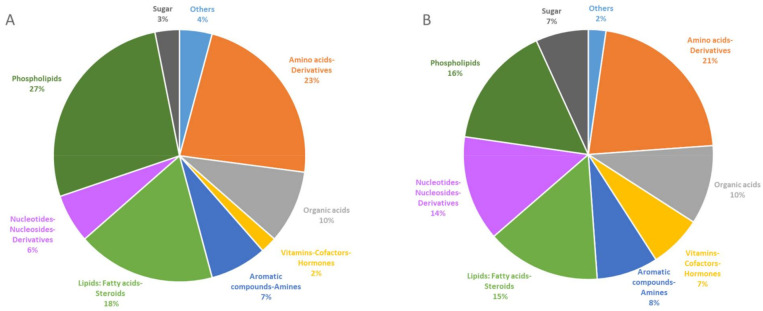
Pie chart visualization of chemical affiliation of metabolites in serum (**A**) and soleus (**B**). VIP metabolites, 96 from serum and 88 from soleus, were classified into nine different chemical classes: phospholipids in dark green, lipids: fatty acids—steroids in clear green, carbohydrate in black, amino acids-derivatives in orange, organic acids in grey, vitamins—cofactors–hormones in yellow, aromatic compounds—amines in dark blue, nucleotides—nucleosides derivatives in violet and others in clear blue. Results are given in the percentage of VIP belonging to the categories.

**Figure 4 metabolites-12-00556-f004:**
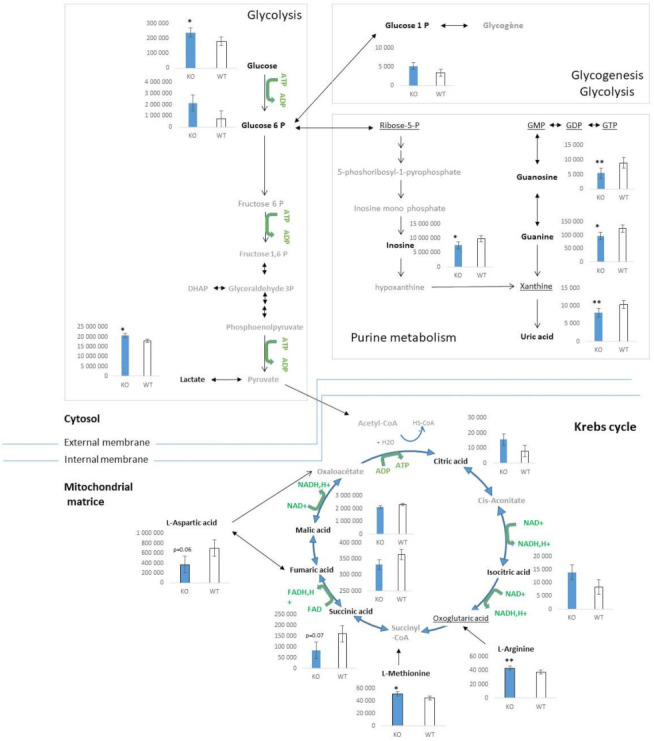
Metabolite changes related to energetic metabolism in soleus of *Klf10* KO mice. Metabolites are represented in their respective metabolism position in glycolysis, glycogenesis—glycolysis, purine metabolism in cytosol and Krebs cycle in mitochondria compartment. Relative metabolite changes shown in the graphs were HPLC-MS results. Open bars, WT mice and filled bars, *Klf10* KO mice. In black bold, metabolites detected and determined as variables with importance in projection (VIP). In black underlined, metabolites were detected but not determined as VIP. In gray, metabolites were not detected. DHAP: dihydroxyacetone 3-phosphate; GMP: Guanosine monophosphate, GDP: Guanosine diphosphate, GTP: Guanosine triphosphate. Values are expressed as mean ± SEM in arbitrary units. * *p*-value < 0.05 and ** *p*-value < 0.01.

**Figure 5 metabolites-12-00556-f005:**
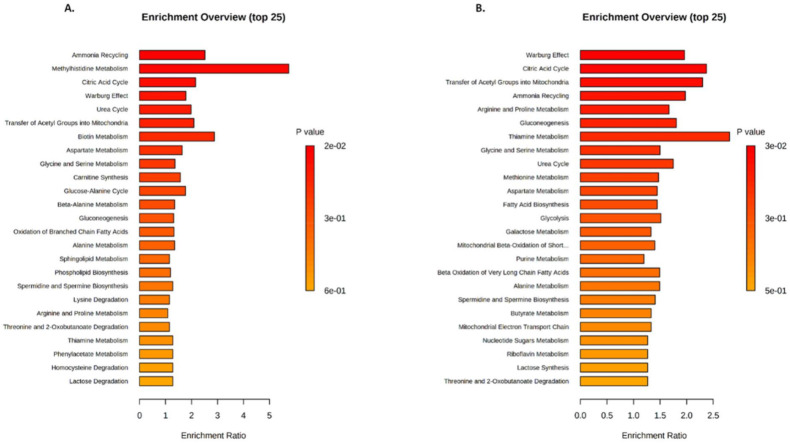
Metabolite Set Enrichment pathway Analysis (MSEA) in serum (**A**) and soleus (**B**). Metabolite set enrichment diagrams (25 top pathways) were obtained using MetaboAnalyst 5.0 on the 96 VIP found in serum and the 88 VIP found in soleus.

**Figure 6 metabolites-12-00556-f006:**
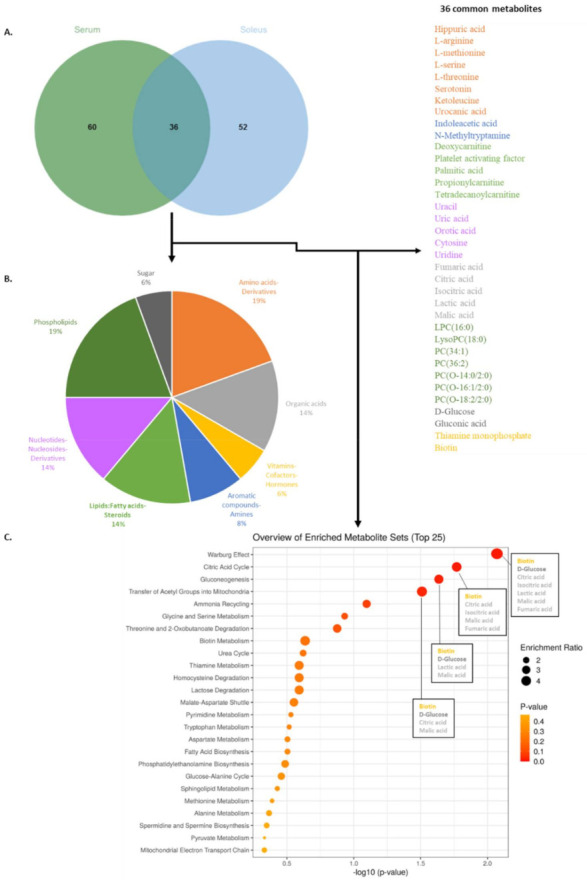
Venn diagram (**A**), pie chart of chemical metabolite repartition (**B**), and MSEA of the 36 metabolites common between serum and soleus (**C**). (**A**) Venn diagram using the 96 VIP from serum and the 88 VIP from soleus were generated and found 36 common metabolites across the two matrices. A list of the common metabolites found both in serum and soleus is given on the right panel of the figure. (**B**) Pie chart using the 36 common metabolites found in the Venn intersection assigned compounds into chemical classes. (**C**) Overview of enriched metabolites sets (bubble plot) that shows the 25 top pathways related to the common 36 metabolites identified both in serum and soleus. For the first four pathways, name of common VIP is in the square plot, colors refer, respectively, to their chemical category. *p*-value is given by red color spot intensity.

**Figure 7 metabolites-12-00556-f007:**
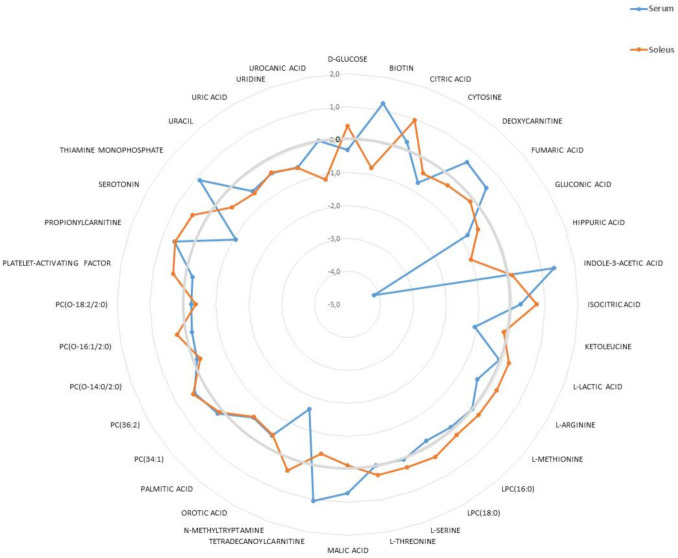
Radar diagram of the 36 VIP found to be common in serum and soleus. Metabolites showed up or downregulation following the bold grey circle giving the zero-base line. Values are given according to log2(FC) of metabolites in serum (blue line) and soleus (red line), respectively.

**Figure 8 metabolites-12-00556-f008:**
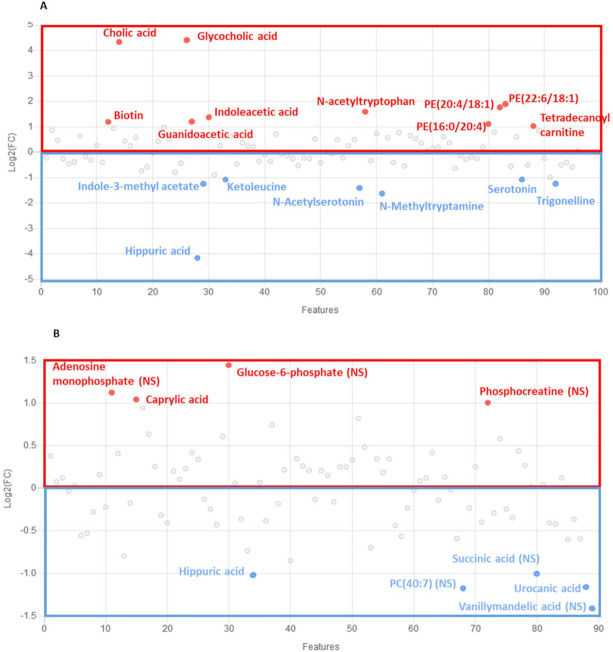
Metabolites as potential biomarkers of serum (**A**) and soleus (**B**) in *Klf10* KO vs. WT mice. Metabolites are selected with a criteria of 2-fold change in regulation and *p*-value < 0.05. NS identify metabolites with two-fold increase and no significant *p*-value. In red are metabolites upregulated, and in blue are metabolites downregulated.

**Figure 9 metabolites-12-00556-f009:**
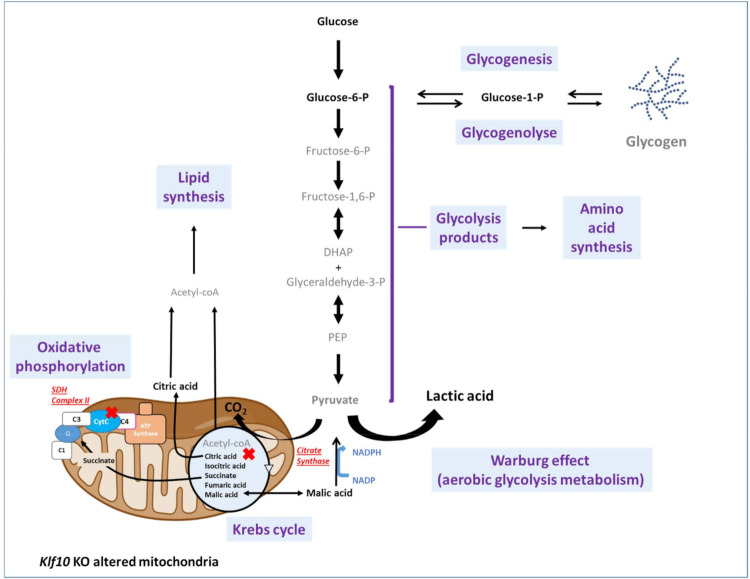
Model of metabolic pathways changes occurring in soleus of *Klf10* KO. A summary of the biochemical pathways related to glucose metabolism in soleus. Metabolites that changed between *Klf10* KO and wild-type mice are indicated in black bold. Red crosses are the metabolic situation and the name of the two enzymes implicated in the described deterioration of mitochondria functions in *Klf10* KO mice (28). DHAP: Dihydroxyacetone phosphate; PEP: Phosphoenolpyruvate; SDH complex II: Succinate dehydrogenase.

## Data Availability

Data are contained in the Appendix A.

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
