# Peer review of "Serum and Soleus Metabolomics Signature of *Klf10* Knockout Mice to Identify Potential Biomarkers"

_metabolites, 2022, doi:10.3390/metabo12060556_

Round 1
Reviewer 1 Report
Baroukh et al.’s manuscript titled “ Serum and soleus metabolomics signature of Klf10 knockout mice to identify potential biomarkers” investigated the critical role of klf10 in metabolism and disease via analyzing the serum and soleu metabolites with or without Klf10 deficient conditions. These findings revealed that (1) the serum and soleus metabolites profiles of wild-type (WT) and knockout (KO) animals vary. In addition, Klf10-deficient mice showed changes in metabolites linked to energy metabolism. (2) The Warburg effect, citric acid cycle, gluconeogenesis, and transfer of acetyl groups into mitochondria pathways are all implicated in the metabolic changes seen in Klf10 KO mice, according to VIP studies. Generally, this study is straightforward, and the results are well presented.
One major question regarding this study.
For the upregulated and downregulated metabolites in the serum and soleus, I would like to see their common and unique up and down metabolites as well as the followed pathway analysis. For example, there are four sub-groups (wt vs. ko).: serum-up, serum-down, soleus-up, and soleus-down. The author should create a Venn diagram to show the common and unique metabolites in each sub-group, as well as evaluate and discuss their connected pathways.
Author Response
Please see the attachement

Reviewer 2 Report
In the current study, the authors analyzed the serum and soleus metabolomics of klf10 knockout mice to identify potential biomarkers for energy related diseases. The authors have studied the amino acids, Krebs cycle intermediates, lipids and phospholipids to elucidate the metabolic alterations in klf10 KO mice. The study is clearly and well written; the introduction is well-justified, the experiments and the data are appropriately analysed and systematically presented. The authors found interesting biomarkers such as cholic acid and glycocholic acid, and interpreted possible related physiological processes. The study presented support that Klf10 is highly involved in the regulation of energy metabolism.
Comments:
1. Why have the authors used female mice for the research?
2. Soleus and EDL represent slow twitch and fast twitch respectively, both are important to the energy metabolism in skeletal muscles. Why have the authors choose just soleus for analysis?
3. Did the authors test the fasting blood glucose of the mice? The glucogenic amino acids are altered, which might cause changes of gluconeogenesis.
4. Glycogen content in soleus could be added in figure 4 to help recognize correlation of metabolites and glycogen metabolim.
5. The alterations in metabolites showed that deletion of klf10 caused changes of glycolysis, Krebs cycle and oxidative phosphorylation in mitochondria, especially the acylcarnitines were increased. The increased carnitines could help transport more fatty aicds into mitochondria for oxidation. It is wondered whether the RER differed between WT and klf10 KO mice?
